# Breast Cancer-Delivered Exosomal miRNA as Liquid Biopsy Biomarkers for Metastasis Prediction: A Focus on Translational Research with Clinical Applicability

**DOI:** 10.3390/ijms23169371

**Published:** 2022-08-19

**Authors:** Oana Baldasici, Valentina Pileczki, Daniel Cruceriu, Laura Ioana Gavrilas, Oana Tudoran, Loredana Balacescu, Laurian Vlase, Ovidiu Balacescu

**Affiliations:** 1The Oncology Institute “Prof. Dr. Ion Chiricuta”, Department of Genetics, Genomics and Experimental Pathology, 400015 Cluj-Napoca, Romania; 2Department of Pharmaceutical Technology and Biopharmaceutics, “Iuliu Haţieganu” University of Medicine and Pharmacy, 400347 Cluj-Napoca, Romania; 3Department of Molecular Biology and Biotechnology, “Babes-Bolyai” University, 5–7 Clinicilor Street, 400006 Cluj-Napoca, Romania; 4Department of Bromatology, Hygiene, Nutrition, “Iuliu Hatieganu” University of Medicine and Pharmacy, 23 Marinescu Street, 400337 Cluj-Napoca, Romania

**Keywords:** breast cancer, metastasis, miRNA, exosomes, extracellular vesicles, biomarker, translational research, clinical applicability

## Abstract

Metastasis represents the most important cause of breast cancer-associated mortality. Even for early diagnosed stages, the risk of metastasis is significantly high and predicts a grim outcome for the patient. Nowadays, efforts are made for identifying blood-based biomarkers that could reliably distinguish patients with highly metastatic cancers in order to ensure a closer follow-up and a more personalized therapeutic method. Exosomes are nano vesicles secreted by cancer cells that can transport miRNAs, proteins, and other molecules and deliver them to recipient cells all over the body. Through this transfer, cancer cells modulate their microenvironment and facilitate the formation of the pre-metastatic niche, leading to sustained progression. Exosomal miRNAs have been extensively studied due to their promising potential as prognosis biomarkers for metastatic breast cancer. In this review, we tried to depict an overview of the existing literature regarding exosomal miRNAs that are already validated as potential biomarkers, and which could be immediately available for the clinic. Moreover, in the last section, we highlighted several miRNAs that have proven their function in preclinical studies and could be considered for clinical validation. Considering the lack of standard methods for evaluating exosomal miRNA, we also discussed the challenges and the technical aspects underlying this issue.

## 1. Introduction

In 2020, GLOBOCAN indicated that breast cancer (BC) had become the leading cause of global cancer incidence worldwide, accounting for 11.7% of all cancers. Among women’s cancers, BC accounts for 25% of new cases and about 17% of deaths [1]. BC represents a highly heterogeneous disease with specific molecular, histological, and clinical features [2]. Gene expression profiling has enabled the molecular portraying of breast cancer which led to the PAM50 classification of breast tumors into five intrinsic subtypes: Luminal A and normal-like (characterized by ER+/PR+ and Ki67 low and good-to-intermediate prognosis), Luminal B (differentiated from Luminal A by high Ki67 and a decline in the patient’s prognosis), HER2+ (characterized by HER2 amplification and the lack of ER and PR), and TNBC (characterized by the lack of receptors ER−, PR−, and HER2−), with both the HER2+ and TNBC subtypes predicting a poor outcome [3]. While the clinical classification based on the expression of estrogen and progesterone receptors (ER, PR) and human epidermal growth factor receptor-2 (HER2) has utility in the selection of targeted therapies, the short-term patient responses and long-term survival remain difficult to predict [4]. In addition, each BC subtype possesses its own proliferation and metastasis capabilities, further hindering the patient’s outcome [5]. 

Despite new insights on molecular subtypes and the improvement of therapy management [6], about 90% of BC deaths are due to metastases [7,8]. BC metastasis represents a multistep evolutionary process involving a variety of cellular and molecular mechanisms. Shortly, this process is depicted as a series of sequential events by which: tumor cells leave their primary site, enter the lymphatic or blood stream, arrive by organotropism in specific secondary sites, and seed and colonize these sites by evading immune surveillance and acquiring resistance to therapeutic intervention [9]. Nevertheless, the molecular mechanism that underlies this process is very complex [10,11,12] and not yet fully elucidated. 

Moreover the clinical–molecular landscape of BC metastasis has revealed two types of metastasis, referred to as de novo BC metastasis (dnBCM) and recurrent metastatic breast cancer (rMBC) [13]. DnBCM is identified in patients that present with stage IV disease at the time of diagnosis, while rMBC appears during tumor evolution after a while in patients initially diagnosed with locally advanced BC. The dnBCM has an incidence of about 6–10% of BC at the point of diagnosis, regardless of the improved screening programs [14]. A TCGA data analysis revealed significant clinical, pathological, and molecular differences between these types of BC metastasis, indicating that dnMBC patients are more responsive to the treatment, with better survival than rMBC patients [15,16]. Although difficult to predict, early diagnosed recurrences in BC patients have a 17–20% chance of improved survival [17].

Consequently, current data regarding the genomic drivers and phenotypic heterogeneity for each clinical BC subtype [18,19] were used to develop several recurrence gene expression-based predictors, including Oncotype Dx, MammaPrint, MapQuant D, Endopredict, the Breast Cancer Index (BCI), and PAM50-ROR [20]. Although gene expression tests have proven their usefulness for predicting the risk of recurrence of ER-positive BC patients, they have limitations in assessing the temporal heterogeneity of cancer and monitoring its evolution [21]. Therefore, identifying the phenotype alterations responsible for tumor evolution remains a challenging task.

A liquid biopsy represents the ideal method for investigating the dynamic BC phenotype of patients in order to identify their risk of relapse and the early detection of systemic dissemination. Previous data have pointed out the role of circulating tumor cells (CTCs), tumor-derived extracellular vesicles, circulating miRNAs, circulating tumor DNA (ctDNA), tumor-associated proteins, and tumor-educated blood platelets as promising and valuable biomarkers for a BC prognosis [22,23]. While CTCs and ctDNA are mostly associated with a liquid biopsy, they have some limitations in both sensitivity and specificity, especially for early cancer diagnostics [24]. 

A promising new class of BC biomarkers is arising in the form of miRNAs transported by tumor-delivered exosomes (miRNA-TDEx) [25,26]. MicroRNAs (miRNAs) are small, non-coding RNAs that regulate gene expression at the post-transcriptional level, being considered master modulators of the cancer phenotype. Therefore, miRNAs are involved in all the hallmarks of cancers, having oncogenic (oncomiR) or suppressor (tumor-suppressor miR) potential. The exosomes are small vesicles of approximately 100 nm, secreted by all cell types, with a role in local and systemic cell-to-cell communication. BC-derived exosomes are essential supporters of BC metastasis by modulating the phenotype of recipient cells and playing an important role in setting the pre-metastatic niche [27,28].

As aberrant miRNA expression is associated with the cancer phenotype, the identification of aberrantly expressed miRNAs in a liquid biopsy could be used to monitor the patient’s cancer status for the early detection of systemic recurrence. The discovery of new predictive miRNA-TDEx biomarkers in a liquid biopsy is essential for improving the management of breast cancer patients by periodical monitorization. Therefore, this paper focuses on identifying miRNA-TDEx with a proven role in BC metastasis prediction that could become useful in clinical practice. Moreover, considering the lack of standard methods for evaluating exosomal miRNA, we also discuss the challenges and the technical aspects underlying this issue. Therefore, we performed a thorough research of the literature presenting the potential role of exosomal miRNAs as biomarkers for metastasis prediction in breast cancer, up to May 2022. The PubMed database was queried using the following keywords: “miRNA exosomes breast cancer metastasis” and “miRNA extracellular vesicles breast cancer metastasis”. From the extensive list of retrieved papers, we focused on reviewing the research articles that evaluated aberrantly expressed tumor-derived exosomal miRNAs in metastatic breast cancer. The papers reporting miRNA evaluation from whole blood lysates or other types of vesicles, as well as small RNAs secreted by stromal or other cell types were excluded. The detailed search strategy is depicted in Figure 1.

## 2. Clinically Validated Exosomal miRNAs as Biomarkers for Breast Cancer Metastasis by Site

Breast cancer subtypes display different particularities with regard to therapy approaches, invasiveness, and survival rate. Moreover, each subtype develops a different metastatic pattern, with specific lesion rates in the secondary organs (Figure 2). Up to 70% [29] of metastases develop in the bone, which is the preferred niche for hormone receptor-positive (HR+) cancers [5]. Of all metastatic sites, bone metastasis tends to predict a better prognosis for the patient in terms of overall survival [30,31]. The liver represents the second most common metastatic site, affecting between 40–50% of all cases, and it is mostly associated with HR− and Her2+ breast cancers [32]. Patients with TNBC have the highest rate of lung and brain metastases [33,34]. Although brain metastasis accounts for the smallest percent of metastatic events in all types of breast cancers, 10–30% [33], more and more cases are being characterized with these particular lesions that unfortunately bare the grimmest overall survival rates [30,31]. Even though the metastatic disease is typically defined as the cancer spread to distant organs, the local progression of breast cancer to the lymph nodes represents a critical event that is indicative of a worse prognosis for the affected patients [35]. Even in the early stages, 10–26% of breast cancer patients diagnosed with T1 tumors present lymph node metastases which have been proven to be correlated with a higher risk of distant spread [36].

MicroRNAs are key regulators of all the hallmarks of cancer [37,38] and their role as potential biomarkers for breast cancer progression and therapy response has also been reviewed elsewhere [39]. However, we believe it is noteworthy to highlight the importance and the potential use of exosome-encapsulated miRNAs in the clinical setting. In the following section, we will present a set of exosomal miRNAs that were identified in liquid biopsy samples of breast cancer patients and were associated with metastatic events at different sites (Figure 2). As their biological functions were also validated in experimental settings, we consider that these miRNAs might represent potential biomarkers that could open new avenues for the early detection of breast cancer metastasis with minimal discomfort for the patient (Table 1). 

### 2.1. Exosomal miRNAs Associated with Lymph Node Metastasis


**MiR-363-5p**


Lymph node metastasis (LNM) represents an important independent risk factor for BC prognosis [40]. The precision of identifying patients prone to develop LNM could influence the accuracy of their treatment. Herein, the potential use of circulating exosomal miRNAs in the detection of lymph node metastasis (LNM) was investigated by Wang et al. [41] in a study comprising 10 breast cancer patients and 10 matched healthy controls. In the breast cancer group, all cases were Luminal-like ER+ and HER2− where 4 patients presented with LNM while the other 6 were without LNM. The microRNAs expression was evaluated in the plasma exosomes of the breast cancer patients and the controls as well as in the tumors and adjacent tissue samples. Of the aberrantly expressed microRNAs, miR-363-5p was significantly associated with breast cancer *(p* = 0.047) and axillary lymph node metastasis (*p* = 0.019). The overexpression of miR-363-5p was observed in both the plasma exosomes and tissue samples of the breast cancer patients when compared to the controls. Interestingly, miR-363-5p was significantly lower in the LN-positive patients than in the LN negative, and its expression in the plasma exosomes was correlated with the expression in the tissue samples. The results were validated in a TCGA cohort which indicates that these observations were specific for ER+ breast cancer. MiR-363-5p had a high LNM detection power with an AUC (area under curve) of 0.958 in the patients’ samples and an AUC of 0.733 in the TCGA dataset. In terms of overall survival (OS), cancer patients with low miR-363-5p expression had significantly lower survival rates (HR = 0.63, 95% CI 0.45–0.89; *p* = 0.0075, log-rank test) than those with high miR-363-5p expression. Moreover, LN-negative patients with low miR-363-5p expression at diagnosis presented a worse outcome (HR = 0.23, 95% CI 0.09–0.60; *p* = 0.00094, log-rank test). In a multivariate survival analysis, a high miR-363-5p expression level was identified as a protective prognostic marker of breast cancer survival (HR = 0.58, *p* = 0.043). In vitro studies demonstrated that the overexpression of miR-363-5p in the MCF-7 breast cancer cell line significantly suppressed its migration, invasion, proliferation, and colony formation capabilities by downregulating the PDGFB expression. Altogether, this data supports miR-363-5p as a reliable candidate as a clinical biomarker. However, testing it in a larger patient cohort might strengthen its applicability. 


**MiR-370-3p**


The biomarker value of miR-370-3p in breast cancer was evaluated by Mao et al. [42] in a cohort of 28 breast cancer patients and 28 matched healthy controls. MiR-370-3p was overexpressed in the tumor tissues, serum, and serum exosomes of the breast cancer patients and was significantly associated with larger tumors, advanced TNM stage, and lymph node metastasis. MiR-370-3p presented a breast cancer diagnostic ability that was specific to the sample origin: for tissue, the AUC was 0.7534, while for the serum and serum exosomes, the AUC values were 0.6735 and 0.6797, respectively. Moreover, miR-370-3p presented higher expression levels in breast cancer cells, especially in highly invasive cell lines. The induced overexpression of miR-370-3p in the MCF-7 and MDA-MB231 cell lines enhanced cancer cell migration and proliferation in vitro and tumor growth in vivo by activating the NF-κB signaling pathway by its FBLN5 target. The inhibition of miR-370-3p had opposing effects, supporting the functional role of miR-370-3p in experimental settings, complementing the clinical observations, and highlighting its role as a tumor suppressor miRNA.


**MiR-222**


Ding et al. [43] identified miR-222 as elevated in the plasma exosomes of breast cancer patients compared to healthy controls. Moreover, they pointed out that a higher level of miR-222 is correlated to lymph node metastatic BC, suggesting that exosomal miR-222 may contribute to the malignancy of BC. To explore the miR-222 mechanism, they conducted in vitro studies on BC cell lines. They identified a significant miR-222 overexpression in the exosomes delivered by MDA-MB-231, considered a high-metastatic cell line, compared to those delivered by MCF-7, a low-metastatic cell line. In addition, miR-222 inhibition reduced the migration and invasion potential of MDA-MB-231 cells. On the contrary, when upregulated in the MCF-7 cell line, it had the opposite effect, activating cell invasion and migration. Interestingly, they observed a change in the phenotype when the exosomes of one cell line were transferred to the culture medium for the other cell line. Thus, they proved that highly invasive cancer cells could transfer exosomal miRNAs to recipient cells, supporting metastasis. Moreover, while searching for the miR-222 mechanism, they identified PDLIM2 as a potential target, an NF-κB inhibitor, and validated this mechanism in both in vitro and in vivo models. In conclusion, miR-222 exosomal expression could be a new indicator of BC lymph node metastasis. However, validation on a large batch of patients would be necessary before its use in clinical practice.


**MiR-148a**


The clinical significance of miR-148a was investigated in a cohort of 125 breast cancer patients, 50 patients with benign tumors, and healthy controls [44]. A quantitative qRT-PCR analysis revealed that breast cancer patients had the lowest expression level of miR-148a in serum exosomes compared to normal subjects and benign patients, and it was associated with lymph node metastasis, tumor differentiation, and TNM stage. Exosomal miR-148a could discriminate between patients with breast cancer and normal controls with an AUC = 0.897 (95% confidence interval = 0.840–0.939, specificity = 80.0%, and sensitivity = 84.0%). Moreover, the miR-148a levels fluctuated in response to cancer treatment, with an initial increase followed by a significant drop, discriminating patients with relapse. Moreover, lower serum exosomal miR-148a was associated with a shorter 5-year OS (*p* = 0.0232) and disease-free survival (DFS, *p* = 0.0103) of breast cancer patients. Together with the TNM stage (HR = 2.863, 95% CI = 1.293–4.822, *p* = 0.010) and lymph node metastasis (HR = 2.051, 95% CI = 1.024–3.028, *p* = 0.019), miR-148a proved to be an independent prognostic factor for breast cancer (HR = 2.460, 95% CI = 1.165–3.620, *p* = 0.015). In breast tumor tissues, a low miR-148a level was correlated with lymph node metastasis and poorer patient prognosis [45,46]. The molecular mechanisms modulated by miR-148a during the metastatic progression were investigated in cell cultures and animal studies. The overexpression of miR-148a in breast cancer cell lines suppresses in vitro migration and invasion by directly targeting WNT-1 and inhibiting the activation of the Wnt/β-catenin pathway [46]. Likewise, through WNT-1 inhibition, the miR-148a overexpression decreased the metastatic potential and lung colonization of murine 4T1 and human MDA-MB-231 cells in vivo [45], supporting the role of miR-148a as a breast cancer metastasis biomarker for the clinic. 


**MiR-3662, miR-146a, and miR-1290**


Li et al. [47] evaluated the potential biomarker value of miR-3662, miR-146a, and miR-1290 in the serum exosomes of 60 breast cancer patients and healthy controls. Their results showed a significant correlation between increased levels of miR-3662, miR-146a, and miR-1290, and lymph node metastasis and advanced stages of the disease (II/III) [47]. In another study, the clinical relevance of miR-1290 was indirectly evaluated in tumor tissues by detecting its target, NAT1. NAT1 detection by IHC was positively associated with the increased OS and DFS of breast cancer patients, especially in those with lymph node metastasis [48]. However, the exact mechanism of action still needs to be elucidated. For miR-146a, experimental studies pointed out its role in supporting breast cancer cells metastasis. Exosomes derived from the highly invasive MDA-MB-231 cell line overexpressed miR-146a and enhanced the metastatic potential of breast cancer cells in vivo [49]. It was proposed that the underlying mechanism is based on TXNIP targeting and the activation of Wnt/β-catenin signaling, which promotes the activation and transformation of normal fibroblasts into cancer-associated fibroblasts. Wnt-β-catenin signaling is also regulated by miR-3662. Mechanistically, miR-3662 targets HBP1 to drive β-catenin accumulation and the activation of Wnt-β-catenin signaling in breast cancer cells, promoting their proliferation, migration, and colony formation capabilities in vitro [50]. 


**MiR-188-5p**


The expression of miR-188-5p was evaluated both as free circulating and exosome encapsulated in the serum of 45 patients with breast cancer, 40 patients with breast fibroadenoma, and 50 healthy subjects [51]. Intriguingly, when increased, the levels of free-circulating miR-188-5p were associated with breast tumors, positive LNM status, and advanced TNM stages, while decreased exosomal miR-188-5p was found in breast cancer patients when compared to the control. Moreover, the expression of miR-188-5p was decreased in the exosomes of the highly metastatic MDA-MB-231 breast cancer cell line when compared to the less invasive MCF-7. The overexpression of miR-188-5p in MDA-MB-231 by miRNA mimics reduced its migration and colony formation capabilities, while the inhibition of miR-188-5p in MCF-7 enhanced the cell colony formation and migration. IL6ST was investigated as an miR-188-5p target, and it was shown to regulate breast cancer cell tumorigenicity in vitro. These results draw attention to the potential role of miR-188-5p as a biomarker for breast cancer lymph node metastasis. However, more studies are needed for elucidating the mechanism of miR-188-5p release into the bloodstream and its clinical application. 

### 2.2. MiRNAs Associated with Bone Metastasis


**MiR-21-5p**


MiR-21 is a highly oncogenic microRNA that functions as an anti-apoptotic and pro-survival factor in multiple types of cancers [52]. In breast cancer, it was consistently found as overexpressed and associated with metastatic progression [53,54], larger tumors, and the presence of circulating tumor cells [55]. Recently, Yuan et al. [49] confirmed the role of miR-21 in breast cancer metastasis, as exosomal cargo, sent to prepare the bone metastatic site. In a cohort of 51 breast cancer patients, miR-21 had the highest expression levels in the serum exosomes of patients with bone metastasis than those with localized disease (n = 21) or with other sites of relapse (n = 9). Experimental studies showed that exosomal miR-21 secreted by cancer cells functions as a critical mediator for establishing a favorable pre-metastatic niche by promoting osteoclast activity and bone lysis both in vitro and in vivo. One molecular target of miR-21 is PDCD4 which serves as an inhibitor of osteoclast differentiation [56]. Besides demonstrating the role of miR-21 in promoting breast cancer bone metastasis, the authors also underlined that their data brings clinical insights for the therapeutic inhibition of miR-21 in osteoclasts that could further be considered for translational medicine. 


**MiR-218-5p**


The bone pre-metastatic niche can also be shaped by exosome-transported miR-218, through its targets COL1A1, YY1, and INHBB that regulate the collagen deposition by osteoblasts. In their in vitro studies, Liu et al. [57] demonstrated the role of cancer cell-secreted exosomal miR-218 in shifting the balance of bone tissue homeostasis toward an osteolytic environment that favors metastatic colonization. The clinical value of miR-218 was investigated in a cohort of 47 stage IV breast cancer patients, with or without bone metastasis. By employing small RNA sequencing, Liu’s group identified that miR-218 significantly upregulated in the sera of breast cancer patients with bone metastasis, compared to those without. MiR-218 was not differentially expressed in other sites of metastasis, indicating a bone metastasis specificity, and a potential biomarker value. 

### 2.3. MiRNAs Associated with Brain Metastasis


**MiR-576-3p and miR-130a-3p**


Curtaz et al. [58] explored the differences in the exosomal miRNA expression and their prognosis significance in terms of the metastatic site between different groups of breast cancer patients. Serum exosomes were isolated from 65 breast cancer patients: primary cancer (n = 15), visceral metastases (n = 18), bone metastases (n = 16), and cerebral metastases (n = 16) as well as 18 healthy age- and sex-matched donors. Their miRNA profiling study reveals that miR-576-3p was significantly upregulated and miR-130a-3p was significantly downregulated when comparing the exosomes of patients with cerebral metastases (AUC: 0.705 and 0.699) with the control group. Moreover, when they compared cerebral metastases to all other metastasis (bone and visceral) and healthy persons and those with primary breast cancer, miR-576-3p was significantly higher (*p* = 0.012, AUC: 0.705, SD 0.071, 95% CI 0.566–0.844). In the same way, miR-130a-3p maintained its significance (*p* = 0.012, AUC: 0.699, SD: 0.060, 95% CI 0.582–0.816). About 80% of the samples from the breast cancer patients with cerebral metastases presented upregulation of miR-576-3p and downregulation of miR-130-3p. However, when investigating the usefulness of these miRNAs as predictors for brain metastases compared with other metastatic sites and primary breast tumors, only miR-576-3p remains statistically significant (*p* = 0.048, AUC: 0.666, SD 0.077, 95% CI 0.516–0.816). MiR-576-3p, by blocking its targets PD-L1 and Cyclin CD1, could maintain immune tolerance within the tumor microenvironment and promote cell cycle progression. However, its mechanism in BC metastasis has to be elucidated. The molecular function of miR-130a was experimentally investigated by Kong et al. [59]. They identified miR-130a-3p downregulated in blood exosomes and tissues from 40 breast cancer patients, being associated with advanced TNM stage (*p* = 0.0014) and lymph node metastasis (*p* = 0.0019). MiR-130a-3p was also found to be downregulated in the breast cancer cell line MCF-7. Promoting miR-130a-3p overexpression inhibited the proliferation of breast cancer stem cells (BCSCs) by inducing G0/G1 arrest and decreased their migration and invasion in vitro by targeting RAB5B. A previous study supported these results as Rab5 proteins have been associated with axillary lymph node metastasis in breast cancer patients [60]. 


**MiR-181c**


An important step of breast cancer metastasis to the brain is the migration of breast cancer cells through the blood–brain barrier (BBB). Tominaga et al. [61] evaluated the role of breast cancer exosomes in mediating the passage of breast cancer cells through the BBB by releasing miR-181c. Their results from 56 BC patients revealed that miR-181c was significantly upregulated in the serum and serum exosomes of those with brain metastasis and also validated the data in highly invasive breast cancer cells with brain tropism. The transfer of exosome miR-181c from BC cells promoted the destruction of the BBB in vitro by targeting PDPK1 in brain endothelial cells and disrupting actin filament organization. Moreover, the EV secretion by BC cells promoted metastatic cells extravasation through the BBB to the brain parenchyma. In vivo, exosomes derived from highly metastatic breast cancer cells with brain tropism preferentially accumulated within the mice brains, promoting a greater permeability of brain blood vessels and contributing to cerebral metastasis. In conclusion, if validated in larger cohorts, the expression of exosomal miR-181c could improve BC prognosis in relation to brain metastasis. 

### 2.4. MiRNAs Associated with Distant Metastasis, without Organ Specificity 


**MiR-105**


MiR-105 was identified as a potential biomarker for breast cancer metastasis by Zhou et al. [62], with significantly higher levels in plasma exosomes of patients (n = 16) that developed metastatic lesions during an average of 4.2 years of follow-up compared to those without metastases (n = 22). The miR-105 expression level was strongly correlated between the plasma exosomes and the tumor tissue, and it was negatively correlated with ZO-1 expression, both in tumor samples and adjacent vasculature. ZO-1 represents a tight junction protein, also known as zonula occludens, and it is crucial for establishing the close connections between the endothelial cells lining the blood vessels. Experimental studies demonstrated that cancer cell-secreted miR-105 downregulates ZO-1 expression, therefore destroying the barrier function of the endothelial monolayers and favoring the trans-endothelial invasion of cancer cells in vitro. Moreover, in vivo experiments demonstrated that exosomes containing high levels of miR-105 accumulated in the brains and lungs of animal models enhanced vascular permeability by targeting ZO-1 and promoted metastasis. They observed a high correlation between the tumor and exosomal levels of miR-105 (r = 0.85, *p* < 0.01). A negative correlation between miR-105 and its target ZO-1 (r = −0.48, *p* = 0.03) and between exosomal miR-105 and tumor-adjacent vascular ZO-1 expression (r = −0.49, *p* = 0.04) was also observed. These results support the functional associations of miR-105 and ZO-1 with breast cancer metastasis and may represent clinical biomarkers for monitoring breast cancer progression as well. 


**MiR-200c and miR-141**


Tumor progression is regulated by a vast number of signaling molecules and transcription factors, blocking or promoting metastasis, depending on the microenvironment signals. FOXP3 (forkhead box P3) is a tumor suppressor gene that regulates the transcription of proto-oncogenes and tumor suppressor genes to exert its anti-cancer function [63]. As seen in a study by Zhang et al. [64], under various circumstances, as pleiotropically functional molecules secreted by cancer cells, miR-200c and miR-141 presented variable expression levels with regard to sample provenience. As downstream targets of FOXP3, miR-200c and miR-141 were downregulated in the breast cancer tissue of animal models and TCGA human breast cancer tissue samples but upregulated in plasma samples of both humans and mice with metastatic disease. High levels of circulating miR-200c and miR-141 were associated with tumor metastasis in a cohort of 259 human subjects, including 114 patients with breast cancer, 30 patients with benign breast tumors, 21 women with a family history of breast cancer, and 94 healthy women. The power of discriminating between distant metastatic disease and localized cases was predicted with the AUC for plasma miR-200c at 0.770 and 0.678 for miR-141. MiR-200c and miR-141 were identified at high levels in the exosomes from breast cancer cell lines culture media, indicating their cancer cell origin; however, due to their different expression levels between the primary tumor and plasma samples, the exact mechanism involved in tumor progression still needs to be elucidated.


**MiR-7641**


In a study by Shen et al. [65], miR-7641 was identified by microarray miRNA profiling, comparing the exosomes secreted by metastatic MDA-MB-231 and non-metastatic MCF-7. Then, miR-7641 was validated for both endogenous and exosomal expression by qRT-PCR and selected as important for clinical investigation. Further, miR-7641 was evaluated in a cohort of 28 breast cancer patients, of which 13 patients presented distant metastasis while the other 15 were diagnosed with localized disease. The exosomes isolated from the plasma of patients with metastatic progression revealed high levels of miR-7641 when analyzed by qRT-PCR. The role of miR-7641 was functionally investigated in breast cancer cell lines and animal models by modulating its exosomal expression and evaluating the biological effects induced by their different up-loading. Their data demonstrate that miR-7641 [65] upregulated in exosomes increases cell proliferation, migration, and invasion, while miR-7641 downregulated in exosomes has the opposite effect, concluding that BC-derived exosomal miR-7641 can be transferred to distant recipient cells, sustaining BC invasion and metastasis.

**Table 1 ijms-23-09371-t001:** Validated exosomal miRNA biomarkers useful for translational research with clinical applicability.

Nr crt	miR	Function	Metastatic Site	Number of Cases	Clinical Significance	Biological Function	Target Gene	Refs.
1	miR-363-5p	Tumor-suppressor miR	Lymph node	10 BC (6 LNM+, 4 LNM−) and 10 healthy controls	Significantly associated with breast cancer (*p* = 0.047) and axillary lymph node metastasis (*p* = 0.019). High LMN detection power with an AUC of 0.958 in patients’ samples and an AUC of 0.733 in TCGA dataset. OS prediction (HR = 0.63, 95% CI 0.45–0.89; *p* = 0.0075, log-rank test).	Suppresses migration, invasion, proliferation, and colony formation in vitro.	PDGFB	[41]
2	miR-370-3p	Oncomir	Lymph node	28 BC and 28 healthy controls	Serum exosome overexpression was associated with larger tumors (*p* = 0.042), advanced TNM stage (*p* = 0.0273), and lymph node metastasis (*p* = 0.0193).	Promotes in vitro proliferation and migration and in vivo tumorigenesis.	FBLN5 and NF-kB signaling	[42]
3	miR-222	Oncomir	Lymph node	38 BC (19 LNM+, 19 LNM−) and 19 healthy controls	Higher expression significantly associated with breast cancer and lymph node metastasis	Promotes in vitro proliferation, migration, and invasion.	PDLIM2 and NF-kB signaling	[43]
4	miR-148a	Tumor-suppressor miR	Lymph node	125 BC, 50 benign tumors, and 40 healthy controls	Low expression correlated with lymph node metastases (*p* = 0.0011), poor tumor differentiation (*p* = 0.0167), and advanced TNM stage (*p* = 0.0004). Breast cancer diagnosis biomarker AUC = 0.897 (95% confidence interval = 0.840–0.939, specificity = 80.0%, sensitivity = 84.0%). Independent prognostic factor for breast cancer (HR = 2.460, 95% CI = 1.165–3.620, *p* = 0.015). Higher expression associated with a longer 5-year OS (*p* = 0.0232) and DFS (*p* = 0.0103).	Suppresses in vitro cancer cell migration and invasion and lung metastasis in vivo.	WNT-1	[43,44]
5	miR-3662, miR-146a, miR-1290	Oncomir	Lymph node	60 BC, 20 healthy controls	Higher expression correlated with lymph node metastasis and later disease stages (II/III).	Promotes proliferation, migration, colony formation in vitro. Sustains tumor growth and metastasis in vivo.	NAT1, TXNIP, HBP1	[46,47,48]
6	miR-188-5p	Tumor-suppressor miR	Lymph node	45 BC, 40 breast fibroadenoma, 50 healthy controls	Exosomal miR-188 downregulated in breast cancer patients. High levels of free-circulating serum miR-188-5p associated with advanced TNM stages and lymph node metastasis.	Inhibits the migration, invasion, and colony formation in vitro.	IL6ST	[51]
7	miR-21	Oncomir	Bone metastasis	51 BC (21 bone metasasis, 21 localized disease, 9 other metastatic sites)	Higher expression correlated with bone metastasis in breast cancer patients.	Promotes osteoclast activity in vitro and the formation of the bone pre-metastatic niche in vivo.	PDCD4	[56]
8	miR-218	Oncomir	Bone metastasis	47 BC (33 bone metastasis, 14 other metastatic sites)	Higher expression correlated with bone metastasis in breast cancer patients.	Inhibits the deposition of collagen in osteoblasts in vitro and promotes the formation of the bone pre-metastatic niche in vivo.	COL1A1,YY1, INHBB	[57]
9	mir-576-3p and miR-130a	Tumor-suppressor miR	Brain	65 BC (15 primary cancer, 18 visceral metastasis, 16 bone metastasis, 16 liver metastasis) and 18 healthy controls	Cerebral metastasis biomarker AUC: 0.699 (*p =* 0.012, SD: 0.060, 95% CI 0.582–0.816).	Not evaluated in this study.	-	[58]
10	miR-130a-3p	Tumor-suppressor miR	Lymph node	40 BC and 40 healthy controls	Lower expression was associated with advanced TNM stage (*p =* 0.0014) and lymph node metastasis (*p =* 0.0019).	Inhibits proliferation, migration, and invasion in vitro.	RAB5	[59]
11	miR-181c	Oncomir	Brain	56 BC	Higher serum and serum exosome expression was associated with brain metastasis (*p* < 0.05).	Promotes blood–brain barrier destruction in vitro and metastatic niche formation in brain in vivo.	PDPK1	[61]
12	miR-105	Oncomir	Not specific		Higher exosome and tissue expression associated with distant metastasis.	Promotes endothelial barrier destruction, trans-endothelial migration, and invasion of cancer cells in vitro and in vivo.	ZO-1	[62]
13	miR-200c, miR-141	Oncomir	Not specific	114 BC, 30 benign tumors, 94 healthy controls	Higher plasma expression associated with breast cancer metastasis. AUC: 0.770 for miR-200c and AUC: 0.678 for miR-141.	Promotes tumor metastasis in vivo.	-	[64]
14	miR-7641	Oncomir	Not specific	28 BC (13 metastatic, 15 localized disease)	Higher expression associated with distant metastasis.	Promotes cancer cell proliferation and invasion in vitro and tumor formation in vivo.	-	[65]

## 3. Technical Aspects of Biomarker Discovery for the Clinic

Identifying exosomal miRNAs with biomarker values for the clinical setting has been a consistent effort in recent years. Some miRNAs were already found to be associated with breast cancer metastasis in patients’ blood samples (Table 1) and represent promising targets for further validation. Other miRNAs (listed in Table 2) were functionally described in experimental settings but were not yet evaluated in the clinic, and they represent interesting subjects for future validation. The available information on exosomal miRNAs is undeniably important as a strong basis for further studies. However, the technical difficulty posed by working with these kinds of samples still constitutes major caveats that must be addressed. Therefore, in the following section, we will discuss the most critical aspects of the technical issues arising while working with exosomal miRNAs and some potential solutions to these problems. 

### 3.1. Sample Processing

Most of the studies included in our review present exosomal miRNAs that were isolated from serum or plasma samples. As part of a liquid biopsy, both serum and plasma can be considered for identifying exosomal miRNA-based biomarkers [66]. A good blood sample quality is highly important because, due to specimen handling, hemolysis may occur and therefore alter the miRNA expression [67]. Consequently, when exosomal miRNAs are used for biomarker investigation, we recommend including only clear serum/plasma samples in the analysis. In general, the available sample volume was relatively low, ranging from 0.2 to 2 mL. However, this cannot be considered a rule, as in half of the studies, no data regarding the starting serum/plasma volume were presented. However, considering the improvements made in the field of modern molecular analysis, only small sample volumes are necessary for evaluation, so the amount of the starting material remains at the latitude of the investigator and the downstream processes intended to follow. For exosome isolation, three methods are frequently considered: ultracentrifugation (UC), precipitation, and column-based isolation. Although UC is seen as the golden standard for exosome isolation, commercially specific precipitation kits represent a reliable and easier alternative. Obtaining a good vesicle sample quality is absolutely important for further analysis. Therefore, the concentration and size consistency of the extracted vesicles is vital [68]. For most of the exosome isolation techniques that are currently accepted, a pre-filter step for input serum/plasma could improve the quality of the samples. While analyzing the methods employed for exosome isolation, we identified only five studies that include a pre-filter step (0.22 µm) for serum/plasma processing, followed by the UC isolation step in two studies and precipitation in three. Based on our experience, we suggest a pre-filtration step to be added in the precipitation-based methods for exosome isolation to ensure that only vesicles with a specific diameter remain in the samples. By performing this step, an increase in the accuracy of the final result is expected to be achieved. Following isolation, several characterization methods can be employed to determine whether the extracted components are exosomes or not. Most of the studies examined for our review use Western blot (WB) to examine the expression of exosome protein markers (CD63, CD9, TSG101, Alix, and HSP70) [27]. In addition, in eight studies, the supplementary characterization of the exosome morphology, size distribution, and quantities was also conducted using transmission electron microscopy (TEM) and a nanoparticle tracking analysis (NTA). The isolation of miRNAs from exosomes was performed by using either commercially available kits in eight studies, or by using the classical extraction, with Trizol/Qiazol/Trizol LS, in four of the studies. These two methods are commonly used when isolating miRNAs from exosomes, favoring either purity or yield, making it difficult to ensure both high yield and great purity at the same time. Therefore, the interchangeable use of different exosome isolation and miRNA extraction methods creates a source of variation between the studies.

**Table 2 ijms-23-09371-t002:** Validated exosomal miRNA biomarkers useful for translational research with clinical applicability.

Nr crt	Exozomal miRNA	Ser/Plasma	Blood Volume	Serum/Plasma Filtration	Exosomes Isolation/Kit	Exosomes Characterization	RNA Extraction from Exosomes	miRNA System	Normalizer	Year	Refs.
1	miR-363-5p	plasma	5 mL	0.22 μm filter	Ultracentrifugation at 100,000× *g*	TEM, NTA, and WB (CD63, TSG101, and calnexin)	miRNeasy Mini kit	miScript SYBR Green	U6	2021	[41]
2	miR-370-3p	serum	250 µL serum	No	Evs precipitation (ExoQuick precipitation solution)	not specified	QIAzol LS	miScript SYBR Green	U6	2021	[42]
3	miR-222	plasma	not specified	0.22 μm filter	Evs precipitation (Exoquick Exosome Isolation Kit)	TEM and WB (CD63, TSG101)	Exoquick Exosome Isolation Kit	SYBR Premix ExTaq reagent	U6	2018	[43]
4	miR-148a	serum	not specified	No	Evs precipitation (Exosome Precipitation Solution)	not specified	miRNeasy Mini kit	miScript SYBR Green PCR Kit	cel-miR-39	2020	[44]
5	miR-3662, miR-146a, and miR-1290	serum	2 mL	No	Evs precipitation (Exosome Isolation Reagent).	TEM, NTA, and WB (TSG101 and CD63)	HiPure Serum miRNA Kit	miDETECT A Track miRNA qRT-PCR Starter Kit	Not specified	2021	[47]
6	miR-188-5p	serum	not specified	0.22 μm filter	Evs precipitation (ExoQuick exosome precipitation solution)	NTA and WB (CD9 and CD63)	miRNeasy Serum/Plasma Kit	miScript SYBR Green	cel-miR-39	2019	[51]
7	miR-21	serum	1 mL	No	Ultracentrifugation at 120,000× *g*	TEM, NTA, and WB (TSG101. Alix, and HSP70)	exoRNeasy Serum/Plasma Midi Kit	SYBR Premix Ex Taq reagernt	cel-miR-39	2021	[56]
8	miR-218	serum	500 µL of serum	No	Ultracentrifugation at 110,000× *g*	NTA	Trizol	miScript SYBR Green	miR-140-3p	2018	[57]
9	mir-576-3p, miR-130a-3p	serum	0.5–1 mL	No	Evs precipitation (Total Exosome Isolation reagent from serum)	ELISA (CD63 and CD9)	Total Exosome RNA and Protein Isolation Kit	TaqMan miRNA Advanced	miR-320	2022	[58]
10	miR-130a-3p	circulating blood	not specified	No	Column-based isolation (ExoQuickExosomal Extraction Kit)	not specified	Trizol	TaqMan MicroRNA Assay kits	U6	2018	[59]
11	miR-181c	serum	not specified	0.22 μm filter	Evs precipitation (Total Exosome Isolation (from serum))	WB (CD9, CD63, cytochrome C)	RNeasy Mini Kit	TaqMan Micro-RNA Assays	U6	2015	[61]
12	miR-105	serum	not specified	No	Ultracentrifugation at 110,000× *g*	TEM	TRIZOL LS for serum exosomes	miScript SYBR Green	miR-16	2014	[62]
13	miR-200c and miR-141	plasma	200 µL	No	Ultracentrifugation at 100,000× *g*	not specified/not performed	miRNeasy Serum/Plasma Kits	TaqMan Micro-RNA Assays	cel-miR-39	2017	[64]
14	miR-7641	plasma	not specified	0.22 μm filter	Ultracentrifugation at 100,000× *g*	TEM and WB (CD9 and CD63)	miRNeasy Serum/Plasma Kit	miScript SYBR Green	U6 and cel-miR-39	2021	[65]

Abbreviations: EVs—extravesicles; TEM—transmission electron microscopy; NTA—nanoparticle tracking analysis; WB—Western blot.

### 3.2. Analytical Platforms

Multiple platforms are used for exosomal miRNA profiling, including next-generation sequencing (NGS), microarray, or PCR array. To strengthen their diagnostic or prognostic value, subsequent validation through quantitative real-time PCR (qRT-PCR) is performed. Ten qRT-PCR miRNA investigations have used SYBR Green with an miRNA-specific primer assay, while the remaining four studies used specific Taqman miRNA assays. Although the Taqman-based miRNA assessment is considered more sensitive than those based on SYBR Green, both approaches are very specific and sensitive, allowing the detection of minimal quantities of miRNAs if the PCR conditions and the primes design are well-assumed [69]. The investigator’s choice depends on their experiences, resources, and the funding allocated. 

### 3.3. Data Normalization

The accurate quantification of exosomal miRNA is faced with certain challenges, concerning relatively low exosomal miRNAs content in biological fluids and the need to identify well-established miRNAs as reference genes. Therefore, there are no standardized methods to evaluate small species of RNAs in liquid biopsies. As no optimal normalization strategy is consensually accepted so far, it leads to variations in the data interpretation and biological predicted effects, limitations in comparing research studies, and misleading conclusions [70]. RNU6B small nuclear RNA (U6), generally used for miRNA tissue normalization, is also used for the data normalization of exosomal miRNAs in many studies. In our analysis, 5 out of 14 studies (Table 2) reported exosomal miRNA normalization to the U6 endogenous housekeeping gene. However, although U6 is considered a stable endogenous gene in tissues, it is not an miRNA molecule and does not reflect the same biochemical features and enzymatic processing. Consequently, recent data suggest that using the same type of RNA species (miRNAs) as normalizers, either as a single miRNA or the average of several miRNAs, may be a more accurate strategy [71]. For a large number of analyzed miRNAs or for experiments with unknown reference genes, the global mean method is considered to be the best approach; however, this is not possible in small-scale studies [72,73,74]. Three articles included in our review have reported the exosomal miRNA expression to miRNA normalizers: miR-320 [58], miR-16 [62], and miR-140-3p [57]. Another approach includes spike-in controls based on exogenous miRNAs. We found four articles [44,51,56,64] that normalized exosomal miRNA expression to exogenous spike-in-like *cel-miR-39*. Normalization to exogenous *cel-miR-39* could be considered a solution for small-scale studies or if no endogenous normalizer could be identified. However, an exogenous normalizer is best used for identifying processing errors and less for determining biological variations in the expression of target microRNAs. Consequently, the normalization just to spike-in exogenous miRNAs could represent a source of bias in providing accurate miRNA expression. From our experience and also considering other published data [75], a combination of an endogenous and an exogenous control miRNA could better solve both non-biological and biological variations underlying exosomal miRNA normalization. From all the studies included in our review, only one paper [65] included a combination of endogenous and exogenous normalizers, indicating that a more standardized methodology is required for enabling the use of exosomal miRNAs as cancer biomarkers

## 4. Perspectives of Exosomal miRNAs from Preclinical Models

Several experimental approaches have been employed for unraveling the mechanisms modulated by exosomal miRNAs during different steps of metastatic dissemination. Therefore, miRNAs represent molecules of interest for biomarker discovery in humans. During each step of the metastatic cascade, tumor cells need to overcome specific barriers posed by crossing through different environments and interacting with host cells. This whole process requires a high level of intercellular communication that is supported by the release of miRNA-loaded exosomes. Due to their capacity for transferring their cargo to tumor cells or other cell types, exosomes are involved in the shaping of the tumor microenvironment and the distant delivery of regulatory factors to other tissues and organs that help create the pre-metastatic niche (Figure 3). 

Migration and invasion are essential prerequisites that allow cancer cells to leave the primary tumor and travel through the stroma in order to reach blood or lymphatic vessels that lead them to distant sites. Locally, exosomes from highly invasive breast cancer cells are able to deliver oncomiRs to normal epithelial and breast cancer cells, leading to an enhanced metastatic potential, by regulating their invasiveness. Experimental studies have demonstrated that exosomes harvested from highly invasive cell lines can transfer miR-370 [76], miR-4443 [77], miR-760 [78], miR-1910 [79], miR-9, and miR-155 [80] to less metastatic tumor or normal epithelial cells, determining an increase in the migration and invasion of the recipient cells and significantly enhancing the metastatic potential of breast cancer cells. 

As tumors grow, so does their need for nutrients and oxygen. By attracting endothelial cells and stimulating angiogenesis, tumors not only gain access to the necessary resources but also create the routes of dissemination. Through miR-210-loaded exosomes, cancer cells stimulate the migration and tube formation of endothelial HUVEC cells in vitro and promote tumor vascularization in vivo [81]. Moreover, exosomal miR-182-5p shows synergic effects in tumor cells and endothelial cells, promoting tumor cells’ motility and endothelial cells’ tube formation, leading to enhanced cancer metastasis [82]. Further, by disrupting the barrier function of endothelial cells in the blood vessels through targeting VE-cadherin, exosomal miR-939 supports tumor cells’ extravasation [83].

Moreover, in order to create an immune microenvironment conducive with tumor progression, cancer cells corrupt the tumor-infiltrating immune cells. The transfer of miR-138-5p to macrophages, by cancer cell-derived exosomes, induced their phenotypic reprogramming from tumor suppressive (M1) toward a tumor supportive state (M2) in vitro and promoted lung metastasis in BALB/c mice. Moreover, the elevated levels of miR-138-5p in the serum exosomes of breast cancer patients were positively associated with later stages, while the tissue expression of its target, KDM6B, was significantly lower in cancer patients than in normal controls [84]. On the other hand, the stimulation of macrophages to secrete pro-inflammatory factors such as IL-1b, IL-6, and TNF-α by exosomal transfer of miR-183 led to tumor growth and metastasis of the 4T1 breast tumor model in vivo [85]. Interestingly, within mice bearing metastatic tumors, serum exosomes displayed a different miRNA profile, with the miR-155 levels significantly increased and miR-205 significantly decreased compared to mice bearing non-metastatic tumors. Experimental studies indicated that miR-155 and miR-205 could regulate tumor growth and metastasis in opposing ways, suggesting the importance of the fine balance in miRNA expression for regulating tumor progression. The mechanism of miR-155-driven metastasis was proposed to be relying on pro-inflammatory cytokine secretion, as high levels of IL-6 and IL-17 were detected in mice bearing highly metastatic tumors and were associated with the exosomal miR-155 expression in these animals [86]. 

The modulation of the immune responses is also vital during the formation of the pre-metastatic niche. By suppressing the local immunity, tumor-derived exosomes can distantly support the colonization of the pre-metastatic site. Let-7 downregulation due to its binding by Lin28 allowed the building of an immune-suppressive pre-metastatic niche in the lungs, by enabling neutrophil recruitment and N2 conversion in vivo [87]. Let-7 restoration in exosomes was able to abolish the immunosuppressive and pro-metastatic effects of breast cancer cell-derived exosomes, highlighting its extensive inhibitory role in tumor progression [87,88]. Another important factor in the development of the pre-metastatic niche is the metabolic reprogramming of the resident cells. By releasing miR-122, cancer cell-derived exosomes downregulate PKM and GLUT1 in neurons, astrocytes, and fibroblasts, therefore reducing the glucose consumption in mice brains and lungs. Subsequently, the increased availability of glucose in these tissues favors the metastatic colonization of cancer cells and tumor spread [89]. In bone, the pre-metastatic niche formation is heavily regulated by miR-19a and miR-20a-5p which promote osteoclast differentiation and proliferation, producing bone lesions and tissue re-modeling [90,91].

Considering their role in shaping cancer progression as demonstrated by the experimental results and, in some cases, in patients’ tissue or blood samples, we believe that these microRNAs listed in Table 3 could represent interesting molecules for future studies on human serum and plasma samples.

**Table 3 ijms-23-09371-t003:** Exosomal miRNAs from preclinical models.

Nr crt	Exozomal miRNA	Regulation	Biological Function in Metastasis	Target	Evaluation	Refs.
1	miR-1910-3p	oncomir	Migration and invasion	MTMR3	in vitro in vivo	[79]
2	miR-370	oncomir	Migration and invasion	-	in vitro	[76]
3	miR-4443	oncomir	Migration and invasion	TIMP2	in vitro in vivo	[77]
4	miR-760	oncomir	Migration and invasion	ARF6	in vitro	[78]
5	miR-9, miR-155	oncomirs	Migration and invasion	PTEN, DUSP	in vitro	[80]
6	let-7a	tumor suppressor	Migration and invasion	c-Myc	in vitro in vivo patients tissue	[88]
7	miR-182-5p	oncomir	Angiogenesis	CMTM7	in vitro in vivo patients tissue	[82]
8	miR-210	oncomir	Angiogenesis	Ephrin-A3	in vitro in vivo	[81]
9	miR-939	oncomir	Extravasation	VE-cadherin	in vitro in vivo patients tissue	[83]
10	mir-155-5p, miR-205-5p	miR-205 tumor suppressor; miR-155 oncomiR	Immune response	IL-6, IL-17	in vitro in vivo	[86]
11	miR-138-5p	oncomir	Immune response	KDM6B	in vitro in vivo patients plasma	[84]
12	miR-183-5p	oncomir	Immune response	PPP2CA	in vitro in vivo	[85]
13	let-7a	tumor suppressor	Immune response	-	in vitro in vivo patients tissue	[87]
14	miR-122	oncomir	Pre-metastatic niche formation	PKM, GLUT1	in vitro in vivo	[89]
15	mir-19a	oncomir	Pre-metastatic niche formation	PTEN	in vitro in vivo patients tissue and plasma	[90]
16	miR-20a-5p	oncomir	Pre-metastatic niche formation	SRCIN1	in vitro	[91]

## 5. Conclusions

During the last decade, numerous research groups have concentrated their efforts on identifying blood biomarkers that could reliably discriminate the patients with aggressive, highly metastatic breast cancers. In this regard, exosomes gained a central spot of interest due to their capacity to transport and deliver miRNAs and induce modifications in the recipient cells. Moreover, the high stability of exosome-encapsulated miRNAs from patients’ blood recommends them as perfect sources of minimally invasive biomarkers. As seen in this review, there are several miRNAs already evaluated in human samples, which still require extensive testing on extended patient cohorts but look promising as future metastasis biomarkers that could lead to improved patient management and more personalized therapeutic schemes. Moreover, miRNAs from preclinical models can bring up valuable information that could open new avenues toward deepening our understanding of the distant intercellular communications and their roles in breast cancer dissemination. 

## Figures and Tables

**Figure 1 ijms-23-09371-f001:**
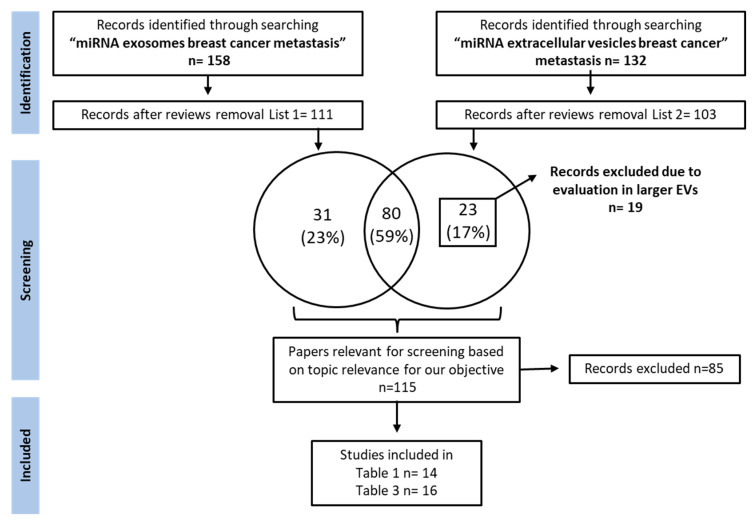
**Study selection flow chart depicting the search strategy used for selecting papers relevant for this review.** A thorough research of PubMed literature was performed. Reviews were excluded from the retrieved results and papers were screened for relevant articles investigating the role of exosomal miRNAs as breast cancer metastasis biomarkers. MiRNAs of interest were grouped by their level of relevance to the clinic into two tables: one containing miRNAs with immediate applicability as liquid biopsy biomarkers (Table 1) and another one containing miRNAs validated in preclinical studies, with potential application to the clinical setting (Table 3).

**Figure 2 ijms-23-09371-f002:**
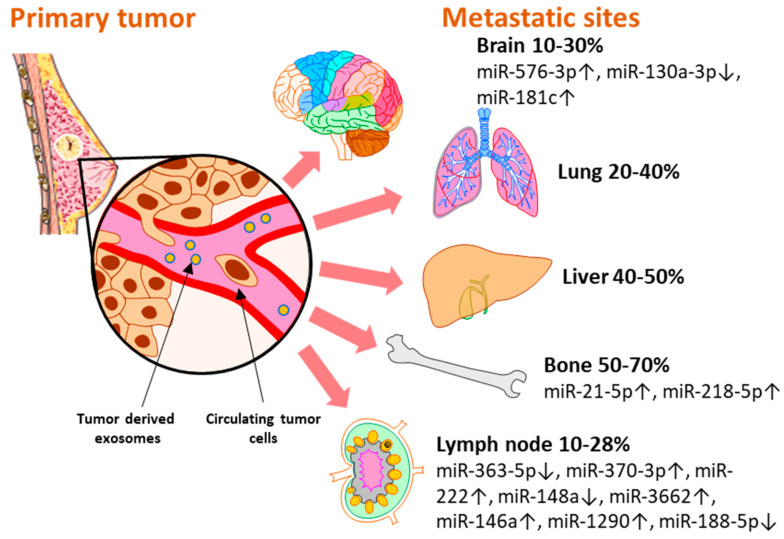
The distribution of miRNAs associated and validated with breast metastasis sites: brain (miR-576-3p, miR-130a-3p, miR-181c), bone (miR-21-5p, miR-218-5p), and lymph node (miR-363-5p, miR-370-3p, miR-222, miR148a, miR-3662, miR-146a, miR-1290, miR-188-5p). Breast cancer has several preferential metastasis sites, with different incidences of secondary lesions. Exosomal miRNAs released by tumor cells have been investigated as blood-based biomarkers for metastasis prediction with regard to the metastatic site. Several miRNAs reported in the literature have been validated in liquid biopsy samples of breast cancer patients and represent potential biomarkers with immediate applicability for the clinic.

**Figure 3 ijms-23-09371-f003:**
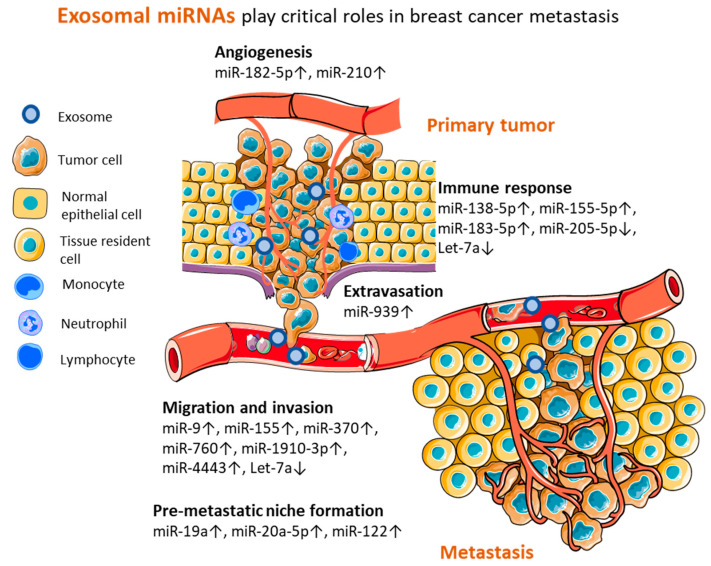
Exosome miRNAs identified by preclinical studies as relevant for breast cancer progression. During the metastatic cascade, exosomal miRNAs released by cancer cells play a pivotal role by regulating key steps, such as angiogenesis, immune evasion, extravasation, migration, and invasion, and the formation of the pre-metastatic niche. In vitro and in vivo studies suggest that these miRNAs could represent potential biomarkers for breast cancer metastasis but require further investigation in liquid biopsy samples.

## Data Availability

Not applicable.

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
