# Peer review of "Breast Cancer-Delivered Exosomal miRNA as Liquid Biopsy Biomarkers for Metastasis Prediction: A Focus on Translational Research with Clinical Applicability"

_ijms, 2022, doi:10.3390/ijms23169371_

Round 1

Reviewer 1 Report

This review is well written and I have only a few comments:

- consider rather than having a data on the search terms in text and figure, to remove the text and just have all the data in the figure

- on page 2 line 87 - there are empty brackets

- figure 2: please add based on which references this figure has been developed

Author Response

Reviewer(s)' Comments to Author:

Reviewer: 1

  1. the title should be changed because doesn't describe correctly the main topic of the manuscript. I suggest to highlight that you are analyzing liquid biopsy strategy.

We agreed with this point of view and we modified the title according to the suggestion:

The new Title: Breast cancer-delivered exosomal-miRNA as liquid biopsy biomarkers for metastasis prediction: A focus on translational research with clinical applicability

  1. The main characteristics of breast cancer and its classification should be deepened. To this purpose the following manuscripts could be useful the doi: 10.4252/wjsc.v11.i9.594 doi: 10.3389/fendo.2018.00492

  • We thank the reviewer for the suggestion and the reference, we added a brief description of breast cancer PAM50 classification in the Introduction section:

BC represents a highly heterogeneous disease with specific molecular, histological, and clinical features [2]. Gene expression profiling has enabled the molecular portraying of breast cancer which led to the PAM50 classification of breast tumors into five intrinsic subtypes: Luminal A and normal like (characterized by ER+/PR+, Ki67 low and good to intermediate prognosis), Luminal B (differentiated from Luminal A by high Ki67 and a decline of the patient’s prognosis), HER2+ (characterized by HER2 amplification, the lack of ER and PR), and TNBC (characterized by the lack of receptors ER-, PR-, HER2-), both HER2+ and TNBC subtypes predicting a poor outcome [3].

  1. Authors indicated the strategy applied to revise current literature data. The search should be performed accordingly to PRISMA guideline. - instead of inclusion exclusion criteria I suggest the PICOS criteria

  • We understood from our reviewers that introducing a methods section in a literature review can be misleading. Our review was not intended to be a Systematic Review, so we believe that PRISMA methodology doesn’t apply in this case. In order to give our paper a more widespread structure for a literature review, we removed the methods section and described the search strategy at the end of the introduction, followed by a schematic representation (Figure 1) as follows:

Therefore, we performed a thorough research of the literature presenting the potential role of exosomal miRNAs as biomarkers for metastasis prediction in breast cancer, up to May 2022. PubMed database was queried using the following keywords: "miRNA exosomes breast cancer metastasis" and "miRNA extracellular vesicles breast cancer metastasis". From the extensive list of retrieved papers, we focused on reviewing the research articles that evaluated aberrantly expressed tumor-derived exosomal miRNAs in metastatic breast cancer. Papers reporting miRNA evaluation from whole blood lysates or other types of vesicles, as well as small RNAs secreted by stromal or other cell types were excluded. The detailed search strategy is depicted in Figure 1

Reviewer 2 Report

The present manuscript thoroughly described the known exosome-associated miRNA involved in breast cancer progression and metastasis formation, highlighting their potential use as clinical biomarkers. The Authors also paused on the critical points of the implementation of miRNAs as biomarkers, raised by the lack of a consensus in the pre-analytical and analytical methodologies.

The manuscript is well structured and exhaustive in the topic described.

Only few suggestions are made:

- In paragraph 4 where all the single miRNA are described, I suggest to examine in depth the molecular function of each miRNA, not simply indicating the miRNA target

- Figure 2 as well as Figure 3 legends need to be better detailed

- I suggest the Authors to consider a recently published review (doi:10.3390/ijms21082805) on the role of miRNA in breast cancer bone metastasis and their potential role as therapeutic targets.

Author Response

Reviewer(s)' Comments to Author:

Reviewer: 2

  1. Consider rather than having a data on the search terms in text and figure, to remove the text and just have all the data in the figure

  • Thank you for the observation! As also stated before, we made the suggested modifications and added the information regarding the search strategy within the Introduction and the legend of the Figure 1:
  • Therefore, we performed a thorough research of the literature presenting the potential role of exosomal miRNAs as biomarkers for metastasis prediction in breast cancer, up to May 2022. PubMed database was queried using the following keywords: "miRNA exosomes breast cancer metastasis" and "miRNA extracellular vesicles breast cancer metastasis". From the extensive list of retrieved papers, we focused on reviewing the research articles that evaluated aberrantly expressed tumor-derived exosomal miRNAs in metastatic breast cancer. Papers reporting miRNA evaluation from whole blood lysates or other types of vesicles, as well as small RNAs secreted by stromal or other cell types were excluded. The detailed search strategy is depicted in Figure 1
  • Figure 1. Study selection flow chart depicting the search strategy used for selecting papers relevant for this review. A thorough research of PubMed literature was per-formed. Reviews were excluded from the retrieved results and papers were screened for relevant articles investigating the role of exosomal miRNAs as breast cancer metastasis biomarkers. MiRNAs of interest were grouped by their level of relevance to the clinic into 2 tables: one containing miRNAs with immediate applicability as liquid biopsy biomarkers (Table 1) and one containing miRNAs validated in pre-clinical studies, with potential application to the clinical setting (Table 3).

  1. on page 2 line 87 - there are empty brackets
  • It is now corrected, thank you
  1. figure 2: please add based on which references this figure has been developed
  • We moved the figure at the end of the text that opens the chapter on clinically validated miRNAs in order to ensure a better readability and clarify how the miRNAs depicted in Figure 2 were selected, as well as where further information and references can be found.
  • In the following section, we will present a set of exosomal miRNAs that were identified in liquid biopsy samples of breast cancer patients and were associated with metastatic events at different sites (Figure 2). As their biological functions were also validated in experimental settings, we consider that these miRNAs might represent potential biomarkers that could open new avenues for early detection of breast cancer metastasis with minimal discomfort for the patient. (Table 1).

Reviewer 3 Report

My suggestions:

the title should be changed because doesn't descriibe correctly the main topic of the manuscript. I suggest to highlight  that you are analyzing liquid biopsy strategy.

The main characteristics of breast cancer and its classification should be deepen. To this purpose the following manuscripts could be useful the doi: 10.4252/wjsc.v11.i9.594 doi: 10.3389/fendo.2018.00492

_ Authors indicated the strategy applied to revise current literature data. The search should be performed accordingly to PRISMA guideline. 

- instead of inclsuon exclusion criteria I suggest the PICOS criteria

Author Response

Reviewer(s)' Comments to Author:

Reviewer: 3

  1. In paragraph 4 where all the single miRNA are described, I suggest to examine in depth the molecular function of each miRNA, not simply indicating the miRNA target
  • The main biological functions regulated by each miRNA are described in the main text and listed in the corresponding table (1 or 3) and where it was possible, brief information regarding the molecular pathways involved was given. As the reference papers where the functional studies are described do not further investigate the entire signaling cascade and only demonstrates the effect of each miRNA upon its target, we believe that it is more accurate to only refer to these molecules, as each gene can be involved in multiple signaling mechanisms with divergent biological repercussions.

  1. Figure 2 as well as Figure 3 legends need to be better detailed
  • Thank you for the observation, we added more information to the figure legends as follows:
  • Figure 2. The distribution distribution of miRNAs associated and validated with breast metastasis sites: brain (miR-576-3p, miR-130a-3p, miR-181c), bone (miR-21-5p, miR-218-5p) and lymph node (miR-363-5p, miR-370-3p, miR-222, miR148a, miR-3662, miR-146a, miR-1290, miR-188-5p). Breast cancer has several preferential metastasis sites, with different incidences of secondary lesions. Exosomal miRNAs released by tumor cells have been investigated as blood-based biomarkers for metastasis prediction with regard to the metastatic site. Several miRNAs reported by the literature have been validated in liquid biopsy samples of breast cancer patients and represent potential biomarkers with immediate applicability for the clinic.
  • Figure 3. Exosome miRNAs identified by pre-clinical studies as relevant for breast cancer progression. During the metastatic cascade, exosomal miRNAs re-leased by cancer cells play a pivotal role by regulating key steps such as angiogene-sis, immune evasion, extravasation, migration and invasion and the formation of the pre-metastatic niche.In vitro and in vivo studies suggest that these miRNAs could represent potential biomarkers for breast cancer metastasis, but require further in-vestigation in liquid biopsy samples.

  1. I suggest the Authors to consider a recently published review (doi:10.3390/ijms21082805) on the role of miRNA in breast cancer bone metastasis and their potential role as therapeutic targets.
  • Thank you for the suggestion, we read the paper with great interest and considered it as an important resource so we cited and added it to the references. The text presenting this reference is:

MicroRNAs are key regulators of all hallmarks of cancer[37,38- doi:10.3390/ijms21082805] and their role as potential biomarkers for breast cancer progression and therapy response has also been re-viewed elsewhere [39]. However, we believe it is noteworthy to highlight the im-portance and the potential use of exosome encapsulated miRNAs in the clinical setting.

Moreover, we did an additional English language and spell check to ensure clarity across text and improved readability. Also, references were checked for their relevance to the contents of the manuscript.
